# Study on Microstructure and Properties of Mechanically Deposited Zn-Sn Coating

**Peng Liu, Shengmin Wang \*, Chengyu Wang and Xiaojun Zhao**

Faculty of Materials Science and Engineering, Kunming University of Science and Technology, Kunming 650093, China
\* Correspondence: 11303063@stu.kust.edu.cn

**Abstract:** A Zn-Sn coating of ~30 μm thickness was prepared on an iron substrate by mechanical deposition using zinc and tin powders as raw materials. The Zn-Sn coating consists of zinc powder particles physically stacked with tin powder particles and filled with reduced tin, and the tin content in the coating is 20%–30%. The resulting Zn-Sn coating was characterized and analyzed with scanning electron microscopy (SEM), X-ray photoelectron spectroscopy (XPS), X-ray diffraction (XRD), polarization curves (Tafel), electrochemical impedance (EIS), and energy dispersive X-ray spectrometry (EDS). The results showed that the Zn powder was co-deposited with the Sn powder in a portable manner and the Sn powder was deflected and deformed to a great extent. The spot-flocculated reduced Sn also covered the surface of the Zn powder to fill the interstices of the coating to make the coating more compact. Compared with the pure Zn coating, the Zn-Sn coating has a positive shift of 68 mV in the self-corrosion potential in the polarization test, and the corrosion current was only 20% of that of the pure Zn coating. The reduced Sn had a shielding effect on the Zn powder and at the same time, in combination with inert tin powder, the polarization resistance of the plated layer increased to 1118 $\Omega/cm^2$. Furthermore, compared to the pure zinc layer, the time of white rust and red rust increased by 24 and 240 h, respectively. In addition, the XPS results showed that the Zn-Sn plating layer was clearly passivated, which was mainly due to the formation of $Zn(OH)_2$ and $Sn(OH)_2$. The results also emphasized that the tin element in the Zn-Sn plated layer can maintain the morphology of zinc powder, compact the plating layer, and prevent the release of corrosion products, thus improving the corrosion resistance of the Zn-Sn coating.

**Keywords:** Zn-Sn coating; mechanical deposition; corrosion resistance; tin powder; passivation

## 1. Introduction

Zn-Sn coating, with a potential between zinc and tin, has both the sacrificial anode characteristic of zinc metal and the isolation of tin. It also has good resistance to sulfur dioxide atmosphere and chloride ion corrosion in corrosion resistance tests [1–5], and is considered an ideal alternative to toxic lead–cadmium layers. With the development of Zn-Sn coating process technology, compared with electroplating, electroless plating and hot-dip plating, the mechanical deposition has attracted the attention of researchers for its non-hydrogen embrittlement, process flexibility, and easy adjustment of coating thickness [6]. The salt spray resistance of mechanically deposited Zn-Sn layers is mainly determined by the tin content in the coating, and mechanically deposited Zn-Sn coating can exhibit strong protective properties when the tin content is in the range of 20% to 30% [7]. It has been reported that the mechanically deposited Zn-Sn layer is three to four times more resistant to salt water and salt spray corrosion than the mechanically deposited pure zinc layer [8]. At present, it is mainly used for small steel fasteners such as drilled screws, nuts, spring spacers, U-rings, etc., which are particularly effective in service in maritime climates and have been included in the relevant technical standards of some coastal countries [9].

With the barrier function of tin and the cathodic protection of zinc, Zn-Sn alloy coating has excellent corrosion resistance [10,11]. During the corrosion process of Zn-Sn alloys, the

active zinc dissolves preferentially to generate passivated products zinc oxide or $Zn(OH)_2$ to cover the coating and form pitting pits. As the pitting deepens to produce tin oxide or $Sn(OH)_2$ generated from tin, the passivated film composed of Zn-Sn products inhibits the corrosion process, while the process is repeated several times with composition changes for specific passivation due to the non-uniform composition of zinc in the coating [12,13]. Although there are many studies on the corrosion resistance of Zn-Sn alloy coating [14–16], there are few studies on the corrosion resistance of mechanically deposited Zn-Sn coating. Moreover, the difference in corrosion resistance between mechanically deposited Zn-Sn coating and Zn-Sn alloy coating in the actual process is unknown.

Because of this, this paper adopts the mechanical deposition method to prepare Zn-Sn coating and study the microstructure and corrosion resistance of the coating to promote the development of alloy mechanical deposition.

## 2. Experimental Materials and Methods

### 2.1. Coating Specimen Preparation Process

The process mainly includes four parts: pre-treatment, establishment of the base layer, coating thickening, and post-plating treatment. The appearance and particle size of zinc powder and tin powder are shown in Figure 1. The mechanical deposition equipment adopts the JDX50-2 mechanical deposition coating barrel developed by Kunming University of Science and Technology (as shown in Figure 2a). The metal zinc content of zinc powder is more than 97% and that of tin powder is more than 92%. The substrate used for the deposition of Zn-Sn coating was Q235 steel (Fe > 98.576% and C < 0.220%) with a thickness of 2 mm, an outer diameter 25 mm, and an inner diameter 14 mm. The impact medium uses spherical mixed glass beads with particle size of 0.5~5 mm and the activator OP-10, while the accelerator consists of inorganic salts such as $SnSO_4$, KCl, etc. The details about the process stages are given below:

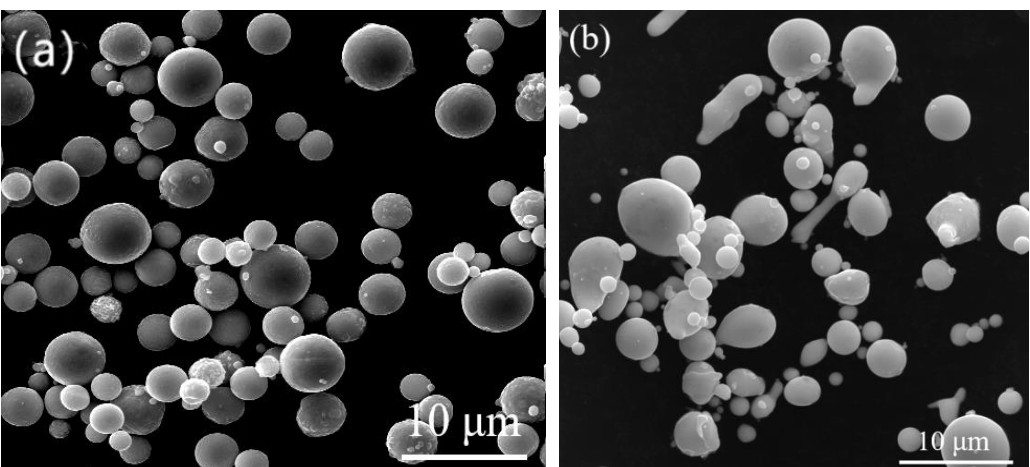

**Figure 1.** SEM images of (**a**) Zn powders and (**b**) Sn powders.

Pretreatment: Alkaline and acidic washing of the substrate is performed using 5% NaOH solution and 15% HCl solution.

Establishing the base layer: Glass beads, spacers, $H_2SO_4$, and $SnSO_4$ are in turn added to the coating barrel to form the tin base layer, and the barrel is rotated for 3 min at a line speed of about 60 m/min.

Coating thickening: Zn-Sn mixed powder and activator with the mass ratio of 19:6 are added for one coating thickening, the rotating speed of the barrel is increased to 100 m/min, and the barrel is rotated for 5 min. Then, Zn-Sn mixed powder and accelerator are added cyclically. After three thickenings are completed, the barrel continues to rotate for 5 min (the process is shown in Figure 2b).

Post-plating treatment: The plated specimens are unloaded, separated, and placed in a box-type resistance furnace for 120 °C + 30 min to dry (the resulting specimens are shown in Figure 2c,d).

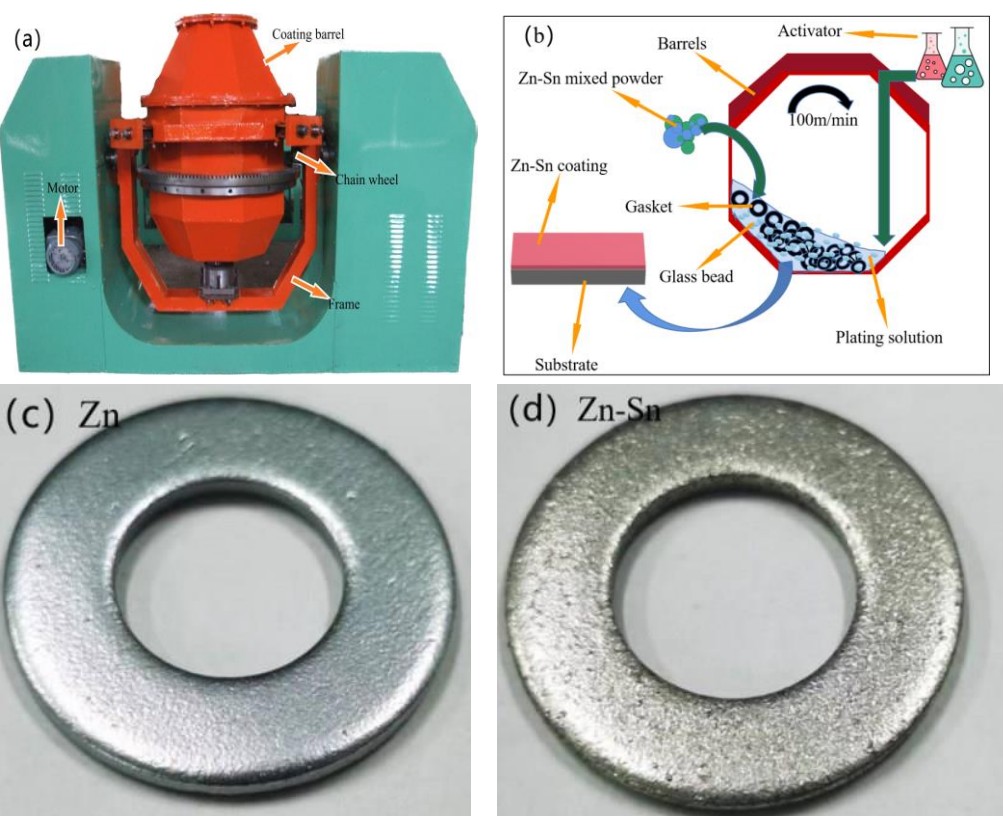

**Figure 2.** Mechanical deposition equipment (**a**); schematic diagram of mechanical deposited process (**b**); surface macrostructure for Zn specimen (**c**) and Zn-Sn specimen (**d**).

## 2.2. Characterization and Analysis of Coating

The polarization curve and impedance spectrum of the analyzed coating were tested by CHI604E electrochemical analyzer (CH Instruments Inc., Shanghai, China). The test sample size was 10 mm × 10 mm, and the scan rate was 1 mV$^{-1}$. A platinum plate was used as the auxiliary electrode and $Hg/Hg_2Cl_2$ as the reference electrode. The corrosion potential (*Ecorr*) and corrosion current density (*Icorr*) were obtained by the Tafel extrapolation. Electrochemical impedance spectroscopy (EIS) measurements were performed at frequencies from 100 kHz to 10 MHz with a sinusoidal signal perturbation of 10 mV, and the fitted parameters were analyzed by Zview software (version 2.4).

A HITACHI-S4800 field emission scanning electron microscope equipped with an energy spectrometer (EDS) was used to analyze the histomorphology and composition of the plated layers.

The composition of the coating was analyzed by a Japanese X-ray diffractometer (D/max-2500, Rigaku corporation, Tokyo, Japan) with a scanning range of 10°~80° and a scanning rate of 10°/min.

The chemical composition and states of the samples were analyzed by XPS (Thermo Scientific K-Alpha Waltham, MA, USA) using an Al Kα X-ray source (1486.6 eV) with charge effect correction using C1s (284.6 eV), with a narrow scan pass energy of 50 eV and with a step size of 0.1 eV. The high-resolution XPS spectra of Zn2p and Sn3d were obtained at a narrow spectral scan pass energy of 50 eV and a step of 0.1 eV.

YH-60B brine spray box was used to test and analyze the salt spray corrosion performance of the coating. The salt solution was 5% NaCl solution, the deposition volume was 1.8 mL/(80 cm²·h), and the test environment temperature was 35 °C. In addition,

the position of the specimen in the salt spray box was changed after 72 h of continuous spraying to eliminate the influence of the position factor on the test, and the time of white rust and red rust of the specimen was recorded.

## 3. Results and Discussion

### 3.1. Analysis of the Microstructure of Zn-Sn Coating

The surface and cross-sectional morphology of the mechanically deposited pure zinc coating and the Zn-Sn coating are shown in Figure 3. The mosaic accumulation of metal particles can be clearly observed on the surface of both coatings. Figure 3a shows the pure zinc surface coating, where the zinc particles of different sizes are intermeshed with each other, with unevenness and a few gaps. Figure 3b shows the Zn-Sn coating, the surface of which is filled by the mosaic accumulation of metal particles (clear Zn-Sn particles are visible in position 3 and in the EDS plot below) and reduced tin (position 7). The denseness of the accumulation is significantly higher than that of the pure zinc coating; the gap between particles disappears and a small number of pores appear on the surface of the coating (position 2). The tin powder particles are non-uniformly distributed and the phenomenon of bias aggregation exists (position 4, 5) to nest with zinc particles (position 1 shows the Zn-Sn partition). The deformation of tin particles by the impact medium is also greater than that of zinc powder (position 4), while the reduced tin is wrapped around the surface of zinc particles (position 6). Figure 3c shows the cross-sectional morphology of pure zinc. It can be seen that the zinc grains are tightly bound in the coating without obvious boundaries and large voids. The deposition mode of tin elements and SEM scanning electron microscopy observation showed that tin elements in the coating are in the form of mosaic and adsorption. The cross section of the Zn-Sn coating was wrapped around the zinc powder surface in the formation of tin ions in tin reduction and formed cotton flocculent on the powder. However, most of the powder particle morphology disappeared, as shown in Figure 3d. EDS was performed in Figure 3b and the results are shown in Figure 4a. The results show that the Zn-Sn coating is mainly composed of pure zinc, pure tin, and related oxides, whose tin content is 24.8% and the oxygen content is 9.9%.

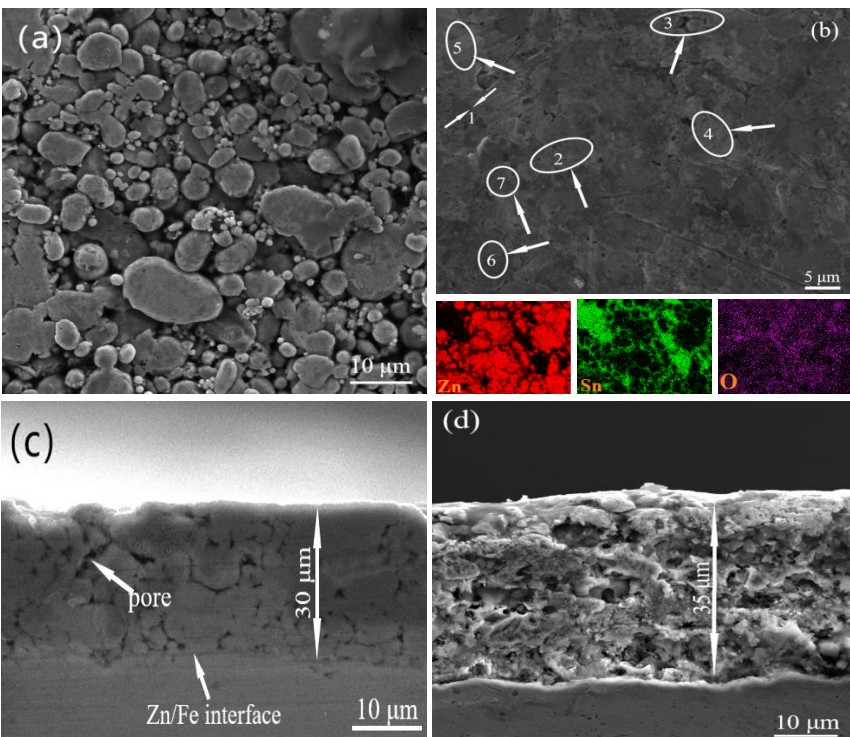

**Figure 3.** Surface morphology of pure zinc coating (**a**) and Zn-Sn coating (**b**); cross-section morphology of pure zinc coating (**c**) and Zn-Sn coating (**d**).

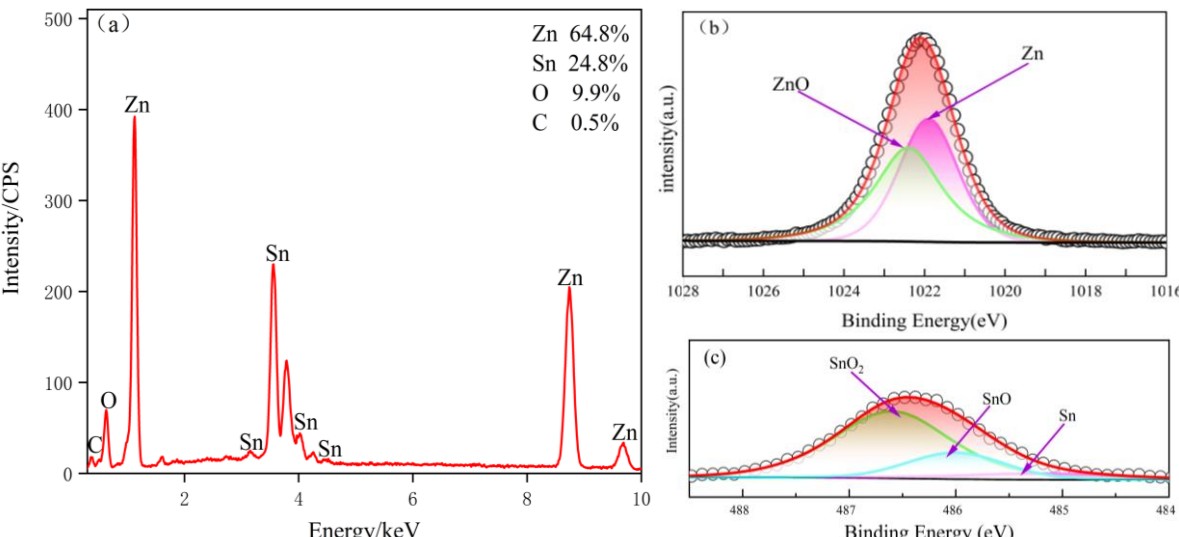

**Figure 4.** EDS energy spectrum of Zn-Sn coating (**a**); XPS core-level spectra of Zn $2p_{3/2}$ (**b**) and Sn $3d_{5/2}$ (**c**) in Zn-Sn coating.

XPS spectra are investigated to reveal the surface state of the Zn-Sn coating. Figure 4b,c exhibits the $Zn2p_{3/2}$ and $Sn3d_{5/2}$ core level spectra of the Zn-Sn coating. In addition, the peaks at 1021.7 and 485.2 eV can be a good fit for Zn and Sn. Due to the natural oxidation of the powder in air, the peaks at 1022.4 and 486.6 eV correspond to ZnO and tin oxide [17], and in addition, the peak at 486 eV is able to match with SnO.

### 3.2. Corrosion Resistance of Zn-Sn Coating

### 3.2.1. Electrochemical Polarization Analysis

The polarization curves of pure zinc coating and Zn-Sn coating in 3.5% sodium chloride aqueous solution are shown in Figure 5, from which it can be seen that the self-corrosion potential of Zn-Sn coating is only positively shifted by 68 mV compared with pure zinc, and there is no significant change in corrosion potential. However, the corrosion tendency of the metal does not completely reflect the actual situation of metal corrosion. The corrosion performance is mainly based on the corrosion rate and the general use of self-corrosion current density to assess the corrosion rate. As shown in Table 1, the self-corrosion current density of Zn-Sn coating is about one-fifth of the pure zinc coating, which indicates that the corrosion tendency of Zn-Sn coating is smaller and better than that of pure zinc coating.

The polarization curves of both the pure zinc coating and the Zn-Sn coating in 3.5% NaCl solution consisted of two passivation zones as shown in Figure 5. The anodic process of the polarization curve was mainly influenced by zinc, and the anodic current increased rapidly, followed by the generation of passivated film by zinc and tin leading to a significant drop in current. In the Zn-Sn coating process, due to the addition of tin powder, the Zn-Sn coating polarization curve in the first passivation zone can detect a passivation platform 68 mV wider than the pure zinc coating, which is related to the tin content in the coating. Thus, the process can be explained by the preferential pitting of the active zinc exposed to the surface coating [18], which is inhibited by the passivated products of zinc with the reduced tin passivated film as the pitting process progresses. In the second passivation zone, the passivation curve of the Zn-Sn coating is smoother and 123 mV wider than that of the pure zinc coating, which may be due to the gradual dissolution of the tin passivated film. The $Sn^{2+}$ activation points on the surface of the tin powder are agglomerated by the traction of the zinc powder and deposited to the substrate, and the agglomerated powder is mechanically impacted and stably embedded on the surface of the substrate. This enables the total content of the tin powder and reduced tin to reach 20%~30% of the coating. In

addition, the oxide film formed by the tin element has a strong physical shielding effect on the zinc powder [19], thus improving the corrosion resistance of the coating.

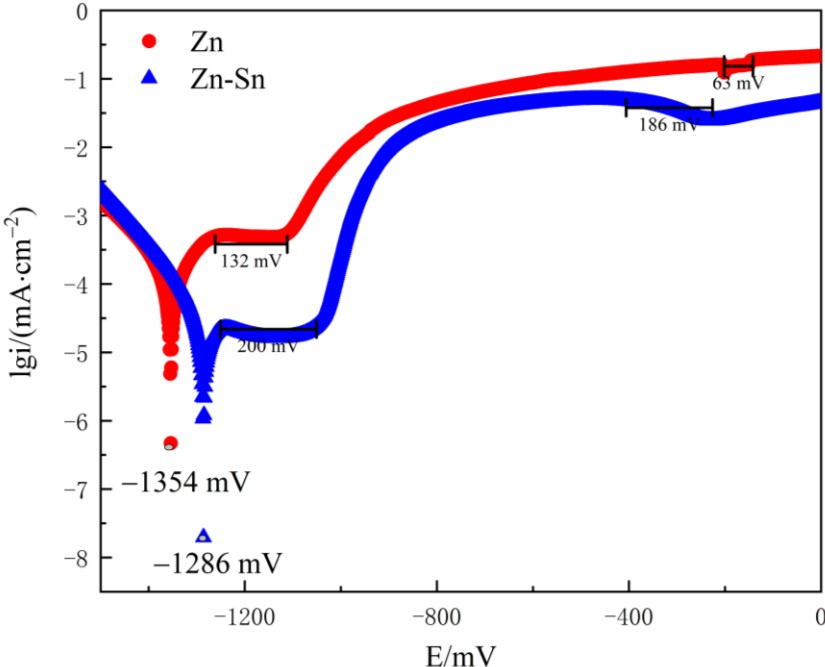

**Figure 5.** Polarization curves of pure Zn and Zn-Sn coatings in 3.5% NaCl solution.

**Table 1.** Parameters of Tafel fitting polarization curve.

| Sample | *Ecorr*/mV | *Icorr*/(µA·cm$^{-2}$) | *Rp*/(Ω·cm$^2$) |
|--------|-----------|------------------------|-----------------|
| Zn | −1354 | $2.35 \times 10^2$ | 375 |
| Zn-Sn | −1286 | $4.51 \times 10^1$ | 1493 |

### 3.2.2. Electrochemical Impedance Spectroscopy

The Nyquist diagrams and equivalent circuits of the pure zinc coating and the Zn-Sn coating are shown in Figure 6. Both the pure zinc coating and the Zn-Sn coating consist of a high-frequency capacitive arc and a low-frequency diffusion arc. The radius of the capacitive arc of the Zn-Sn coating is significantly larger than the radius of the arc of the pure zinc coating, which indicates that the diffusion process of the corrosive medium of the Zn-Sn coating is more inhibited and proves that the Zn-Sn coating has better corrosion resistance. The polarization resistance is derived by fitting the data to an equivalent circuit and shown in Table 2, wherein $R_s$ is the solution resistance, $R_{coat}$ is the coating resistance, $R_{ct}$ is the charge transfer resistance, $Q_{dl}$ is the double layer capacitance at the solution/coating interface, and $CPE_{coat}$ is used instead of pure capacitance to represent the capacitance of the entire coating (including the passivated layer) to match the actual electrochemical process. As shown in Table 2, the resistance of the Zn-Sn coating is significantly greater than that of the pure zinc coating. The passivated film formed on the surface of the coating leads to a Weber impedance at the end of the capacitive arc of the pure zinc coating, which has a shielding effect on the mass transport of reactants and products [20]. So, the charge transfer resistance of the pure zinc coating is smaller than that of the Zn-Sn coating. However, in general, the Zn-Sn coating can maintain a larger radius of capacitive arc resistance due to the protective shielding effect of reduced tin.

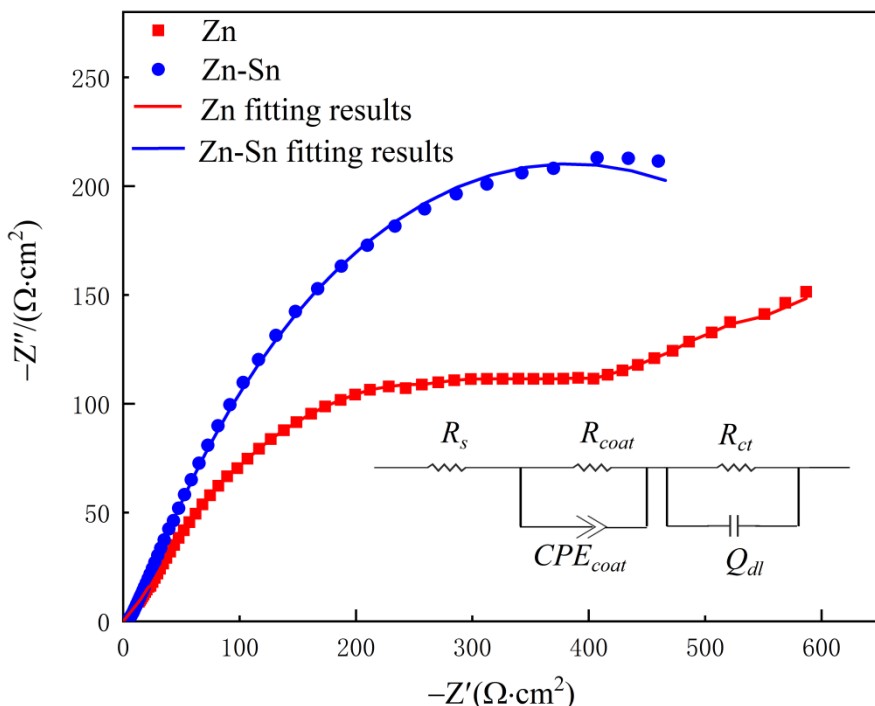

**Figure 6.** Impedance diagrams for Zn and Zn-Sn coating in 3.5% NaCl solution and its equivalent circuit.

**Table 2.** Equivalent circuit fit parameters for the EIS samples.

| Sample | $R_s$/ ($\Omega \cdot cm^2$) | $R_{coat}$/ ($\Omega \cdot cm^2$) | $R_{ct}$/ ($\Omega \cdot cm^2$) | $Q_{coat}$/ ($F \cdot cm^{-2}$) | $Q_{dl}$/ ($F \cdot cm^{-2}$) |
|---|---|---|---|---|---|
| Zn | 5.36 | 509.6 | 207 | $9.03 \times 10^{-2}$ | $6.48 \times 10^{-2}$ |
| Zn-Sn | 4.39 | 1173 | 113.8 | $7.52 \times 10^{-2}$ | $2.21 \times 10^{-2}$ |

### 3.2.3. Neutral Salt Spray Test Analysis

White rust is an oxidation product of zinc, which appears as a white powder and becomes flocculent when it is wet, and red rust is an oxidation product of iron. Therefore, the corrosion progress can be determined by observing the color of the product on the surface of the sample. A neutral salt spray test (the specimen is suspended in the salt spray chamber for continuous spraying) was used to evaluate the corrosion resistance of the Zn-Sn coating and the pure zinc coating. The test results showed that white rust appeared at 48 h and red rust at 120 h for the pure zinc coating with a thickness of 30 μm. However, for the Zn-Sn coating with thickness of 30 μm, white rust appeared in 72 h and red rust appeared in 360 h. This indicates that the salt spray corrosion resistance of the Zn-Sn coating is better than that of the pure zinc coating, and further confirms that the electrochemical test results are consistent with the actual corrosion resistance performance.

The macroscopic morphology of the Zn-Sn coating after 120 h of salt spray treatment is shown in Figure 7a. It can be seen that the white rust is easily agglomerated by external factors, forming striped flocs in the early stage and gradually covering the whole workpiece over time. After 480 h of salt spray treatment, the macroscopic morphology of Zn-Sn coating is shown in Figure 7b. It can be seen that the corrosion of Zn-Sn coating is mainly pitting corrosion. The process is also similar to that of pure zinc coating in the early stages (i.e., the migration of Cl$^-$ enrichment causes activation inside the pores [21]) which accelerates the corrosion of the coating layer. The red rust gradually spreads to the surface, forms a sheet of regional corrosion, and peels off in spots.

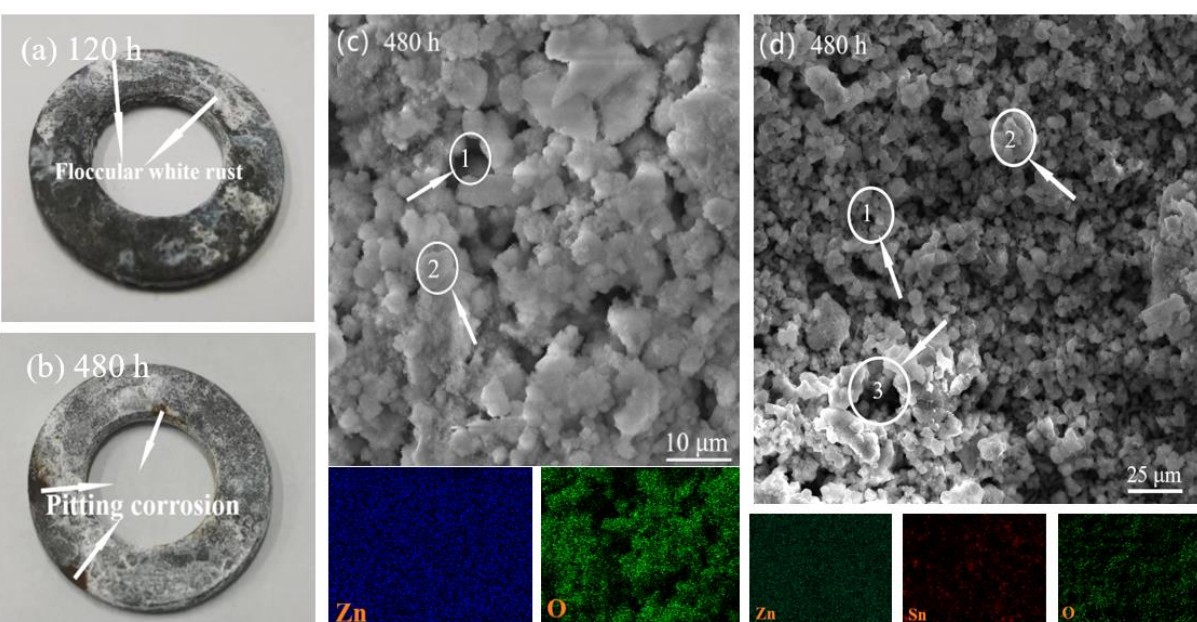

**Figure 7.** Surface macrostructure after salt spray test 120 h (**a**) and 480 h (**b**) for Zn-Sn coating; SEM and EDS image of the pure zinc coating (**c**) and Zn-Sn coating (**d**) after salt spray test 480 h.

The SEM and EDS of the pure zinc coating after 120 h of salt spray treatment are shown in Figure 7c. It can be seen that the swollen and loose white rust occupies the surface of the workpiece, and most of the zinc powder oxidizes into an irregular granular (Position 1) form with more obvious porosity between the particles (Position 2). The SEM and EDS of the Zn-Sn coating after 480 h of salt spray treatment are shown in Figure 7d. It can be seen that the presence of reduced tin not only maintains the morphology of some zinc powder particles (Position 1), but also connects and reduces the fluffiness between the particles (Position 2). So, the coating does not fall off easily after corrosion. Although the porosity of the Zn-Sn coating after corrosion is significantly larger than that of the pure zinc coating (Position 1), the maintenance of the morphology of the zinc powder particles and the isolation of the pitting process by the tin element in the middle and late stages of corrosion make the comprehensive corrosion resistance of the Zn-Sn coating significantly stronger than that of the pure zinc coating.

### 3.3. Passivation Process of Zn-Sn Coating

In order to better explain the passivation process of the Zn-Sn coating, XRD and XPS measurements were performed on the Zn-Sn coating (salt spray) samples after 480 h of salt spray treatment. As shown in Figure 8a, the diffraction peaks of tin in the Zn-Sn coating were significantly enhanced while the diffraction peaks of zinc were significantly weakened due to the coverage of tin elements. The diffraction peaks of zinc and tin in the Zn-Sn coating were sharply weakened or even partially disappeared after the salt spray treatment, and some new peaks appeared. Among them, the $Zn(OH)_2$ diffraction peak was the most prominent, indicating that the corrosion products of the Zn-Sn coating are dominated by $Zn(OH)_2$. The EDS spectral surface scan (as shown in Figure 8b) of the Zn-Sn coating (salt spray) treated with 480 h of salt spray showed that the zinc content on the surface of the specimen decreased from 64.8% before the start of the salt spray test to 58.4% in the time of the 480 h salt spray test. Furthermore, the tin content decreased significantly from 24.8% before the start of the salt spray test to 6.8% at the time of the 480 h salt spray test. The results show that tin is preferentially dissolved by oxidation. From Figure 8b,d, it was found that only flaky or whisker-like reduced tin adhered to the surface of zinc powder particles on the surface of the composite coating after salt spray corrosion, and there were essentially free particle-shaped tin powder attached. This indicates that as the pitting

process deepens, tin powder is affected by the oxidation expansion of zinc powder and salt spray [22] and falls off due to the lack of maintenance force between the zinc powder and the workpiece. However, the bonding force between zinc powder particles by reduced tin can maintain agglomeration and particle morphology. Therefore, in the late stage of coating corrosion, the passivation process of the Zn-Sn coating mainly relies on the shielding effect of the reduced tin and the hindering effect of the zinc powder oxide layer, while in the early stage, it mainly relies on the shielding effect of the tin powder and the reduced tin together with the passivated film of the zinc powder. A small amount of Fe element can be found (as shown in Figure 8b), possibly due to partial exposure of the iron substrate after corrosion or contamination by corrosion of other parts during the salt spray corrosion test.

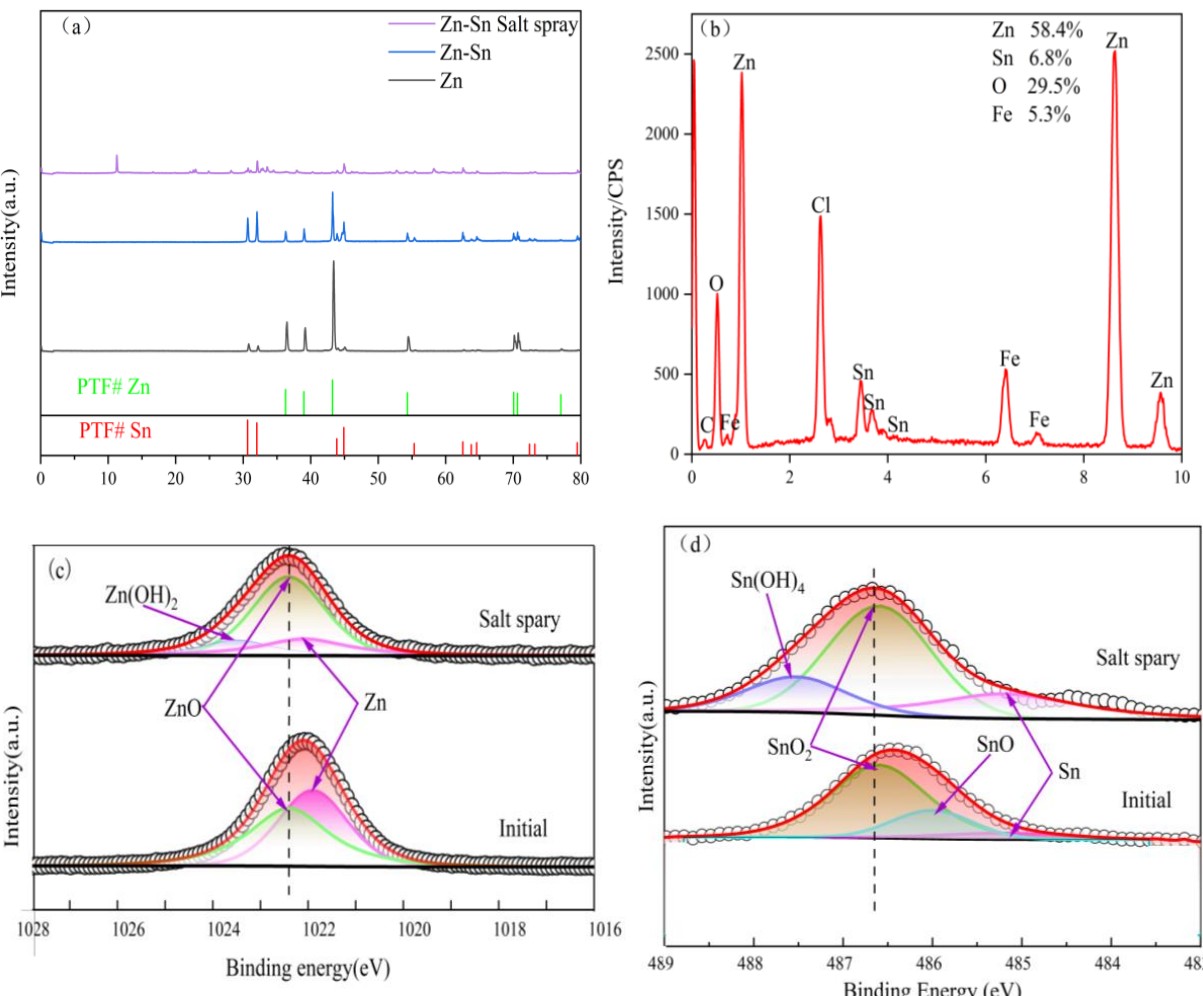

**Figure 8.** (**a**)The XRD map of the Zn and Zn-Sn coating and Zn-Sn coating after 480 h of salt spray treatment; EDS energy profiles of Zn-Sn coating after 480 h of salt spray treatment (**b**); the XPS core-level spectra of Zn $2p_{3/2}$ (**c**) and core-level spectra of Sn $3d_{5/2}$ (**d**).

The evolution of the core spectra of the Zn-Sn coating Zn $2p_{3/2}$ and Sn $3d_{5/2}$ during the neutral salt spray test is shown in Figure 8c,d. After the salt spray treatment of the Zn-Sn coating, it can be observed that the increase in the ratio of oxides of Zn and Sn leads to a shift of the XPS peak position towards high bonding energy [23]. A new peak at 2023.1 eV is observed for Zn, which belongs to $Zn(OH)_2$, and the peaks at 486 and 487.6 eV for Sn can be attributed to SnO and $Sn(OH)_2$ [24], which appears in the coating. The results indicate that the passivation process of Zn-Sn coating depends mainly on the formation of Zn-Sn oxides and hydroxides. The proportion of ZnO in the Zn-Sn coating is significantly higher and some $Zn(OH)_2$ appears after salt spray treatment, and in the Sn $3d_{5/2}$ spectra

the SnO disappeared while the content of $Sn(OH)_2$ and tin oxide increased significantly. This indicates that the continuous generation of Zn-Sn hydroxides facilitates the formation of passivated films and that growth and dissolution occur simultaneously at the interface of passivated film [25,26]. In addition, the corrosion resistance of the coating depends largely on the growth and dissolution rate of ZnO due to the shedding of tin powder particles at the later stages of corrosion.

## 4. Conclusions

In the mechanically deposited Zn-Sn coating, the reduced tin is wrapped around the surface of the zinc powder to shield and maintain the morphology and agglomeration state of the zinc powder, and the tin powder and the zinc powder are mosaicked and meshed with each other, and the tin powder is biased.

Because of the filling of tin element to the coating and the wrapping and shielding of zinc powder particles, the Zn-Sn coating has a positive shift of 68 mV in corrosion potential compared with the pure zinc layer, a decrease of 189.9 $\mu A/cm^{-2}$ in corrosion current density, and a polarization resistance four times that of the pure zinc layer. Therefore, mechanically deposited Zn-Sn plating can effectively inhibit the diffusion of corrosion products and improve the corrosion resistance of the plating.

The passivation process of Zn-Sn coating depends mainly on the formation of Zn-Sn oxides and hydroxides. In the first stage, the reduced tin is preferentially dissolved in the coating, along with the shedding of the tin powder, and in the later stage it relies mainly on the growth and dissolution of the zinc passivation layer.

For the preparation of mechanically deposited Zn-Sn coatings with high tin content, the deposition kinetics of inert metal tin still needs to be investigated in depth. The addition of complex surfactants and mechanical activation can be a future research direction. The activation of tin elements can substantially increase the thickness and the proportion of tin elements in Zn-Sn coating and enhance the corrosion resistance of the coating.

**Author Contributions:** Conceptualization, C.W. and X.Z.; Writing—original draft, P.L.; Writing—review & editing, S.W. All authors have read and agreed to the published version of the manuscript.

**Funding:** This work was supported by the National Natural Science Foundation of China (52161013).

**Institutional Review Board Statement:** Not applicable.

**Informed Consent Statement:** Not applicable.

**Data Availability Statement:** The data that support the findings of this study are available upon reasonable request from the authors.

**Conflicts of Interest:** The authors declare no conflict of interest.

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
