# Peer review of "Study on Microstructure and Properties of Mechanically Deposited Zn-Sn Coating"

_coatings, doi:10.3390/coatings12121919_

Round 1

Reviewer 1 Report

Please see the enclosed file.

Author Response

Thank you for your valuable and thoughtful comments. We have carefully checked in the revised manuscript. Please see the attachment for details of the changes.

Reviewer 2 Report

Liu et al. evaluate the microstructure and properties of mechanically deposited Zn-Sn coatings on iron substrates. The manuscript is well structured and the data shown by the authors support the conclusion, nevertheless its quality can be further improved. Namley, the following aspects can be improved before publication:

1. Authors can also indicate in the Introduction, in what application such Zn-Sn coatings can be used.

2. One question is related to the O amount in the sample (see also comment 3). From EDX, only Zn and S are mentioned (maybe O is from the unknow peak at around 0.5eV). From the fitted XPS peaks of Figure 4a,b both Zn and Sn have oxides (but no O1s peak is shown), - as the authors state, these oxide might be due to natural oxidation. But authors  also have to  include and EDX mapping of O of the top-view in figure 2b or of the Fe/Zn/O in the cross section shown in figure 2d. Or authors can also include a XPS sputter depth oxide (removing some 10-50nm of the top layer), and can evaluate the O amount.

3. Ln 278-293 - authors should also indicate why the Fe is clearly visible in the EDX spectra

4. Overall the figure arrangement can be improved to increase the arrangment of the article and its readability, or additional work needs to be done for the figures. For example:

a) Figure 1 - the SEM images can be on two columns;

b) Figure 2 - similarly, in 2 columns and 2 rows - also the figure legend for each a,b,c,d should be mentioned in the figure caption and not below each figure. Also authors can indicate the Zn/Fe interface in Figuer 2c (as well as some of the gaps which will be filled with Sn), as well as the layer thickness in both Fig 2c,d.;

c) Figure 3 - please use same number of decimals and indicate all peaks on the graph (also the peak around 0.5-0.7 keV

d) Figure 4 - on two columns. Also for each peak fitting plot, the background line does not have to be in the middle of the Y axis, and arrange the plot so that the peaks occupy 70-90% of the figure height. It would also be helpful if authors use some symbols to distinguis between the species and the fitted (envelope) as printed in black and white it would not be really clear.

e) Figure 5, also include a symbol for each polarization curve, to better identify them in black&white print

f) Figure 6 shows only the impedance spectrum (Nyquist) and not the equivalent circuit which is in fig 7. Please merge the figures. Also the the impedance spectra for the Zn, ZnSn coatings should also includethe experimental data and the fitted data.

g) Improve arrangement of Figure 8 - on two columns so that it fits only on 1 page. The details for each figure should be in the figure caption now below each figure. Figure 8a,b, please replace the text into the figure to English and indicate in each figure and caption the time of exposure - 120h and 480h.

h) Figures 9-11 could be merged into one figure, with mentioning all the details into the figure captions. Figure 11a,b - correct text into figure to "Salt spray" - and overall make sure the fittings occupy 70-90% of the figure area.

5. Please check the manuscript from mispelling and minor English corrections: e.g., ln 284 "form" correct to "from"; ln 289: "maintain Therefore," correct to "maintain. Therefore,"; ln 301 correcto to "treatment, the" and so on.

Reviewer 3 Report

This article needs a few  corrections

1.       Chapter 2.1  coating specimen preparation process

Necessary drawings and photographs of the test sample and a schematic diagram of the Zn-Sn coating technology. The presented text description is insufficient.

2.       Chapter 3.1. Analysis of the microstructure of Zn-Sn coating

There are no descriptions in Fig. 2. In Fig. 2, the characteristic areas described in the chapter text are not indicated with arrows. A must to improve.

3.       The reviewer regrets that he does not read in Chinese and cannot read the descriptions in Fig. 8a, b. Please improve on English. In Fig. 8c,d, the characteristic areas described in the chapter text are not indicated with arrows. A must to improve.

4.       Chapter 4. Conclusion . On what basis do you say so, I do not see it in photo metallography

In the mechanically deposited Zn-Sn coating, the reduced tin is wrapped around the  surface of the zinc powder to shield and maintain the morphology and agglomeration state of the zinc powder, and the tin powder and the zinc powder are mosaicked and  meshed with each other and the tin powder is biased.

Round 2

Reviewer 1 Report

Changes are acceptable.

Author Response

Thank you for taking the time to review the manuscript. Your suggestions have important guiding significance for my thesis writing and scientific research work.

Reviewer 2 Report

The authors addressed the points raised in the previous revision. However, considering the modifications, some more minor adjustments can be performed, as detailed below.

1. New figure 2: Figure 2a shows the actual depostion equipment or that of Kunming University of Science and Techology. Additionally, authors can indicate on the photo the major components of the mechanical depositon coating barrel.

2. with respect to answer 4d of previous revision: author can keep their preferred format. If needed, authors can incluide an arrow and indicate the compoenents (e.g., SnO2, Sn, SnO - in Figure 6b, etc.).

Author Response

(The authors gave the same response as above.)
